# Antifungal Activity and Biocontrol Potential of *Simplicillium lamellicola* JC-1 against Multiple Fungal Pathogens of Oilseed Rape

**DOI:** 10.3390/jof9010057

**Published:** 2022-12-30

**Authors:** Wenting Li, Tao Luo, Juncheng Li, Jing Zhang, Mingde Wu, Long Yang, Guoqing Li

**Affiliations:** State Key Laboratory of Agricultural Microbiology, Huazhong Agricultural University, Wuhan 430070, China

**Keywords:** *Simplicillium lamellicola*, endophyte, antifungal activity, oilseed rape, biological control

## Abstract

A fungal strain (JC-1) of *Simplicillium* was isolated from a pod of oilseed rape (*Brassica napus*) infested with the blackleg pathogen *Leptosphaeria biglobosa*. This study was done to clarify its taxonomic identity using morphological and molecular approaches, to characterize its antifungal activity through bioassays and genome-based identification of antifungal metabolites, and to determine its efficacy in inducing systemic resistance (ISR) in oilseed rape. The results showed that JC-1 belongs to *Simplicillium lamellicola.* It displayed a strong antagonistic relationship with *L. biglobosa*, *Botrytis cinerea* (gray mold) and *Sclerotinia sclerotiorum* (stem rot). The cultural filtrates of JC-1 showed a high efficacy in suppressing infection by *S. sclerotiorum* on detached leaves of oilseed rape. Genome analysis indicated that JC-1 has the capability of producing multiple antifungal metabolites, including aureobasidin A1, squalestatin S1 and verlamelin. Inoculation of JC-1 on seeds of oilseed rape caused a suppressive effect on infection by *L. biglobosa* on the cotyledons of the resulting seedlings, suggesting that JC-1 can trigger ISR. Endophytic growth, accumulation of anthocyanins, up-regulated expression of *CHI* (for chalcone isomerase) and *PR1* (for pathogenesis-related protein 1), and down-regulated expression of *NECD3* (for 9-cis-epoxycarotenoid dioxygenase) were detected to be associated with the ISR. This study provided new insights into the biocontrol potential and modes of action of *S. lamellicola*.

## 1. Introduction

*Simplicillium* (*Cordycipitaceae*, Hypocreales and Ascomycota) is a genus established by Zare and Gams in 2001, and the fungi of this genus are characterized by the formation of simple (solitary), unbranched conidiophores (phialides) and short-ellipsoidal conidia in globose slimy heads at the apex of the conidiophores [1]. *Simplicillium* was previously treated as *Verticillium* section Prostrata; therefore, it is closely related to *Verticillium*. In 2001, *Simplicillium* only accommodated three species, namely *S. lamellicola*, *S. lanosoniveum*, and *S. obclavatum* [1]. Since then, the taxonomy of *Simplicillium* has attracted the attention of many researchers, and substantial progress has been made in this aspect. So far, one new variety of *S. lanosoniveum* and at least 18 novel species of *Simplicillium* have been described, including *S. album*, *S. aogashimaense*, *S. calcicola*, *S. chinensis*, *S. cicadellidae*, *S. coccinellidae*, *S. cylindrosporum*, *S. filiforme*, *S. formicae*, *S. formicidae*, *S. humicola*, *S. hymenopterorum*, *S. lanosoniveum* var. *tianjinensis*, *S. lepidopterorum*, *S. neolepidopterorum*, *S. niveum*, *S. scarabaeoidea*, *S. subtropicum* and *S. sympodiophorum* [2,3,4,5,6,7,8,9,10,11]. These *Simplicillium* species have diversified ecological niches, some live with fungi as mycoparasites [1,12,13,14,15,16,17], some live with insects as entomolopathogens [2,3,9,18,19], some live with plants as endophytes or pathogens [20,21,22], and some live as saprophytes in fresh water [7], in soil [8], on animal faeces [11], and even in the deep sea [23].

Previous studies showed several species of *Simplicillium* with the mycoparasitic lifestyle can potentially be used for the control of plant fungal diseases [15,17,19,24,25]. Choi et al. (2008) found that *S. lamellicola* BCP (=*Acremonium strictum* BCP) is a mycoparasite of the gray mold fungus *Botrytis cinerea* [26], and it has been developed as a biofungicide for control of gray mold in tomato and ginseng [25]. Ward et al. (2011) and Gauthier et al. (2014) reported that *S. lanosoniveum* is a mycoparasite of the soybean rust fungus *Phakopsora pachyrhizi*, and it showed a high suppressive efficacy against soybean rust through suppression of the set-up of the populations of *P. pachyrhizi* on soybean leaves [13,24]. Wang et al. (2020) found that *S. obclavatum* is a mycoparasite of the stripe rust fungus *Puccinia striiformis* f. sp. *tritici* (*Pgt*), and it displayed a high efficacy in the suppression of the production of urediniospores by *Pgt* and the germination rates of the *Pgt* urediniospores on wheat leaves [15]. Recently, Zhu et al. (2022) reported that *S. aogashimaense* is a mycoparasite of the powdery mildew fungus *Blumeria graminis* f. sp. *tritici* (*Bgt*), and it had a high efficacy in the suppression of powdery mildew through impairing colony formation by *Bgt* on wheat leaves [17].

Besides mycoparasitism, *Simplicillium* species have other modes of action against plant pathogens, including antibiosis through the production of antimicrobial metabolites [27,28]. Previous studies showed that the BCP strain of *S. lamellicola* can produce verlamelin (a cyclic lipopeptide) and mannosyl lipids [27,28]. Verlamelin showed antifungal activity against many plant pathogenic fungi, including *Alternaria alternata*, *Bipolaris maydis*, *Botrytis cinerea*, *Fusarium oxysporum* and *Magnaporthe grisea*, and this metabolite (100 μg/mL) displayed an effective suppression of the development of barley powdery mildew caused by *B. graminis* f. sp. *hordei* (*Bgh*) on barley leaves. Moreover, the gene cluster (*vlmS*, *vlmA*, *vlmB*, *vlmC*) responsible for the biosynthesis of verlamelin in *Lecanicillium* sp. HF627 (a close relative of *Verticillium*) has been elucidated [29]. Mannosyl lipids had antibacterial activity against two plant pathogenic bacteria, including *Agrobacterium tumefaciens* (crown gall) and *Ralstonia solanacearum* (bacterial wilt) [28]. Another study by Liang et al. (2017) indicated that strain EIODSF 020 of *S. obclavatum* from the deep sea could produce other cyclic peptides with antifungal activity against *Aspergillus versicolor* and *Curvularia australiensis* [23].

Additionally, Zhang et al. (2014), Shen et al. (2012) and Teasdale et al. (2018) isolated *Simplicillium* from plant tissues; the resulting strains of *Simplicillium* were thus considered to be plant endophytes [21,22,30]. However, the endophytic capability of both species of *Simplicillium* has not been experimentally confirmed. Moreover, whether or not these two species can trigger a plant defense response against infection by plant pathogenic fungi remains unknown.

Li (2019) obtained the JC-1 strain of *Simplicillium* from a diseased pod of oilseed rape (*Brassica napus*) infested with *Leptosphaeria biglobosa*, the causal agent of blackleg in oilseed rape [31]. The peculiar association of this strain of *Simplicillium* with *L. biglobosa* prompted us to pose the following two questions: (i) could the JC-1 strain potentially be used to control *L. biglobosa* and other fungal pathogens of oilseed rape (e.g., *Botrytis cinerea*, *Sclerotinia sclerotiorum*)? and (ii) what mechanisms does *Simplicillium* JC-1 own to suppress these fungal pathogens? To address these two questions, we conducted this study to identify *Simplicilliums* JC-1, determine its antifungal activity and capability of producing antifungal metabolites, and detect its efficacy in inducing systemic resistance against infection by *L. biglobosa*.

## 2. Materials and Methods

### 2.1. Fungal Strains and Cultural Media

Four fungal strains were used in this study, including *Simplicillium* JC-1, *Botrytis cinerea* B05.10, *Leptosphaeria biglobosa* W10 and *Sclerotinia sclerotiorum* RAP-1. JC-1 was isolated from a pod of oilseed rape (*Brassica napus* L.) [31]. B05.10 was kindly provided by Prof. Zhonghua Ma of Zhejiang University (Hangzhou, China); it was originally isolated from table grapes (*Vitis vinifera* L.) [32]. W10 was isolated from a diseased plant oilseed rape collected from Wuxue County, Hubei Province, China [33]. RAP-1 was isolated from a sclerotium collected from oilseed rape [31].

Two cultural media, including potato dextrose agar (PDA) and potato dextrose broth (PDB), were used to incubate the fungal strains mentioned above. They were prepared with peeled potato tubers following the routine procedures.

### 2.2. Fungal Identification

The JC-1 strain was identified based on morphological observation and phylogenetic analysis. For morphological observation, mycelial agar plugs (5 mm in diameter) were removed from the colony margin of a PDA culture of JC-1 and inoculated on PDA in Petri dishes (9 cm in diameter), one mycelial agar plug in each dish. The cultures were incubated at 20 °C in the dark for 16 d. Colony morphology (size, color and texture) was observed with the naked eye. To observe the intact conidiophores, a sterilized cover slip (24 × 24 mm, length × width) was inserted into the PDA in front of the colony margin of JC-1 at an angle of ~45°. The culture was incubated at 20 °C until JC-1 grew large enough to reach the cover slip. The cover slip was then carefully taken out, and a drop of lactophenol cotton blue was pipetted onto the cover slip to stain the hyphae, conidiophores and conidia on the cover slip. After staining for 5 min, the cover slip was placed on a glass slide, which was finally placed under a compound light microscope for examination of the morphology of the hyphae, conidiophores and conidia.

In the molecular identification, the JC-1 strain was inoculated on cellophane-film overlays on PDA, and the cultures were incubated at 20 °C for 15 d. The mycelia were harvested using a sterilized aluminum spatula. Genomic DNA was extracted from the mycelial masses using the CTAB method [34]. It was then used as a template to amplify the DNA sequences of the internal transcribed spacer region (ITS), 28S large subunit of rDNA (LSU), small subunit of ribosomal DNA (SSU) and *TEF1α* (for translation elongation factor 1ɑ) with the primer pairs ITS4/ITS5, LROR/LR7, NS1/NS4 and EF1-983F/EF1-2218R, respectively (Appendix A). Details of the PCR reaction system and the thermal program for the PCR were described in Appendix A, respectively. The resulting DNA amplicons were loaded into agarose gels, which were subjected to electrophoresis. After that, the target DNA bands were separately purified from agarose gels, cloned into *E*. *coli* DH5*α* using the vector pMD18-T (TaKaRa Biotechnol. Co., Ltd., Dalian, China), and sequenced. The resulting DNA sequences were submitted to the GenBank public database (http://www.ncbi.nlm.nih.gov, accessed on 5 February 2021) to obtain accession numbers (Table 1). The DNA sequences of ITS alone for 26 fungal taxa and the concatenated sequences of ITS, LSU, SSU and *TEF1α* for fungal 19 taxa (Table 1) were aligned using Clustal W in MEGA X [35]. Phylogenetic trees were inferred based on these two datasets using the procedures described by Guo et al. [36].

### 2.3. Genome Sequencing and Identification of Biosynthetic Gene Clusters for the Secondary Metabolites

The genome-assisted approach was used to determine the capability of producing antifungal metabolites by the JC-1 strain [17]. Genomic DNA was extracted from the mycelial masses of JC-1 using the CTAB method [36]. It was then sequenced at Frasergen Bioinformatics Co., Ltd. (Wuhan, China) using a PacBio Sequel II System and MGISEQ-2000 platform, as well as at Annoroad Gene Technology Co., Ltd. (Zhejiang, China) using the Illumina Novaseq 6000 platform. De novo genome assembly was performed with PacBio long reads using MECAT2 [37]. For correction of the errors, sequence polishing was done with the MGISEQ-2000 and Illumina short reads using the Pilon package v1.23 [38]. Genome annotation was conducted following the pipeline in BRAKER v2.1.5 using the softwares GeneMark-ES v4.68 and Augustus v3.3.3 [39,40,41]. The sequences of the contigs, together with the raw sequencing data, were submitted to the China National GeneBank (CNGB) DataBase with the accession number CNP0003490. The completeness of genome annotation was estimated using BUSCO v4.1.4 with the dataset for fungi in the order *Hypocreales* [42]. The genome was submitted to the website https://fungismash.secondarymetabolites, and the biosynthetic gene clusters for the secondary metabolites (SM) were then searched using antiSMASH v. 6.1.1 [43]. To detect expression of the backbone genes in the SM gene clusters, total RNA was extracted from the JC-1 mycelia in 3- and 9-day-old PDB cultures (20 °C) using Fungal RNA Kit 200 (OMEGA Bio-Tek, Norcross, GA, USA). RNA profiling was done on the MGI-SEQ 2000 platform at Frasergen Bioinformatics Co. Ltd. The resulting reads for the backbone genes were counted as FPKM (fragments per kilobase of exon model per million mapped fragments).

### 2.4. Dual Culturing

To detect the antifungal activity of JC-1, mycelial agar plugs (5 mm in diameter) of JC-1 were inoculated to one side of Petri dishes (9 cm in diameter), each containing 20 mL PDA and one mycelial agar plug (1 cm from the dish rim). Meanwhile, mycelial agar plugs (5 mm in diameter) from fresh PDA were placed in Petri dishes and treated as control. The dishes were placed in an incubator (20 °C) for 10 d, and then, mycelial agar plugs (5 mm in diameter) of each target fungus (*B. cinerea*, *L. biglobosa* and *S. sclerotiorum*) were inoculated on the opposite side of the mycelial agar plug of JC-1 or the control agar plug (1 cm from the dish rim). There were three dishes (replicates) for each dual culture with JC-1 and a target fungus as well as for each single culture with the target fungus alone. The cultures were again incubated at 20 °C until the control dishes for the single cultures were almost colonized by the target fungi. The width of the clear zone between the colonies of JC-1 and a target fungus in each dual culture was measured. The trial was repeated three times.

### 2.5. Determination of the Antifungal Activity of the JC-1 Cultures

Two trials were done to determine the antifungal activity of the PDB cultures of JC-1, with *S. sclerotiorum* as the indicator for the antifungal activity. The first trial was a time-course experiment. Mycelial agar plugs (5 mm in diameter) of JC-1 were inoculated in 250-mL Erlenmeyer flasks, each containing 50 mL PDB, with one mycelial agar plug in each flask; there were 15 flasks in total. The flasks were mounted on a rotary shaker and incubated at 20 °C and 150 rpm for 3, 6, 9, 12 and 15 d. At each time point, three flasks were randomly selected and taken off the shaker. The culture in each flask was centrifuged at 9000 rpm at room temperature for 15 min to obtain the supernatant and the mycelial mass precipitate. The mycelial mass was dried at 50 °C for 48 h and weighed. The pH value of the supernatant was measured using a pH meter. Meanwhile, the supernatant was incorporated into PDA at 10% (*v*/*v*), and in the control treatment, non-utilized fresh PDB was also incorporated into PDA at 10% (*v*/*v*). The PDA of both treatments was loaded separately into Petri dishes, 20 mL per dish, and three dishes (replicates) for each treatment. Mycelial agar plugs of *S. sclerotiorum* were inoculated into the Petri dishes, one mycelial agar plug in each dish. The cultures were incubated at 20 °C for 2 d, the diameter of the colony in each dish was measured, and the percentage of inhibition of mycelial growth was calculated based on the colony diameters in the two treatments [44].

The second trial was a dose-effect experiment, and the 15-day-old PDB cultures of JC-1 were used in this trial. The supernatant was obtained from the PDB cultures by centrifuging, and it was supplemented in PDA by 1%, 2%, 5%, 10%, 15% and 20% (*v*/*v*), and in the control treatment, PDA was supplemented with water alone by 1% to 20% (*v*/*v*). *S. sclerotiorum* was inoculated on PDA with different treatments, and the cultures were incubated at 20 °C for 2 d, colony diameters were measured, and percentages of inhibition of mycelial growth were calculated [44].

Moreover, the basic properties of the antifungal metabolites in the PDB cultures of JC-1 (20 °C, 15 d) were determined, including the therm-stability (40 °C to 121 °C for 10 to 60 min), resistance to UV-irradiation (5 to 100 min), and response to ambient pH (pH 2 to 13 for 24 h). *S. sclerotiorum* was used as the indicator fungus. The details of the experimental procedures for these tests were described in our previous study [45].

### 2.6. Extraction of the Antifungal Metabolites

The procedures to extract the antifungal metabolites from the PDB cultures of JC-1 (20 °C, 150 rpm, 15 d) were outlined in Appendix A. The cultures were centrifuged to remove the mycelial masses. The resulting supernatants were pooled, and acetone was added at a volume ratio of 1:1. The two phases were thoroughly mixed, and the mixture was filtered through four-layered gauze fabric to remove the flocculent metabolites. The filtrate was evaporated in a rotavapor to remove the acetone. Then, ethyl acetate was added to the remaining solution, also at a volume ratio of 1:1, and the two phases were thoroughly mixed three times. After standing at room temperature for 20 min, the mixture was again separated into two phases: the clear organic phase and the turbid water phase. The organic phase was collected with the aid of a separatory funnel and dried by rotary evaporation. The dried extract in the flask was washed with a small amount of methanol, water was added to the resulting solution, and the mixture was freeze-dried. Finally, a yellowish powder was obtained, and it was considered to be the crude extract (CE) of the antifungal metabolites of JC-1.

To determine antifungal activity, the CE was dissolved in water to generate a mother solution at 10 mg/mL, it was supplemented into PDA to the final concentrations of 12.5, 25, 50, 100, 200, 400, or 800 μg/mL, and PDA alone was treated as a control. The PDA for different treatments were separately loaded in Petri dishes, three dishes (replicates) for each treatment, and *S. sclerotiorum* was inoculated on the PDA in these dishes. The cultures were incubated at 20 °C for 2 d, the diameter of the colony in each dish was measured, and the data was used to calculate the percentages of inhibition of mycelial growth [44].

### 2.7. Determination of Control Efficacy of the Antifungal Metabolites

Seeds of oilseed rape (*B. napus* cultivar “Zhongshuang No. 9”) were pre-germinated at 20 °C on moisturized filter papers for 4 d. The germinated seeds were then sown in Organic Culture Mix (Zhenjiang Pei Lei Organic Fertilizer Co. Ltd., Zhenjiang, China, N:P:K = 1:1:1, pH 6) in plastic pots (10 × 10 cm, diameter × height), with five seeds per pot. The pots were maintained in the growth room (20 °C, 12 h light/12 h dark) for 30 d. True leaves (~10 × 8 cm, length × width) were excised from the plants. The detached leaves were randomly selected and placed on moist towels in plastic trays (45 × 30 × 2.5 cm, length × width × height) for the following two assays: (i) the supernatant assay with four treatments, including three treatments with the supernatants of JC-1 from 9-, 12- and 15-day-old PDB cultures (e.g., SU-9d, SU-12d and SU-15d, respectively) and a control treatment with water alone (control); (ii) the crude extract assay with six treatments, including three treatments with the JC-1 CE at 200, 400 and 800 μg/mL (e.g., CE-200, CE-400 and CE-800, respectively), the supernatant treatment with SU-15d of JC-1, the fungicide treatment with prochloraz at 450 μg a.i./mL (e.g., PR-450), and the control treatment with water alone. Prochloraz EW (MIAOLIANG^TM^) was purchased from Henan Guangnong Pesticide Factory (Mengzhou, Henan province, China). The solutions (SU-9d, SU-12d, SU-15d, CE-200, CE-400, CE-800, and PR-450) and water alone were separately amended with Tween 20 at 0.2% (*v*/*v*) and applied onto the leaves in the trays at ~1 mL per leaf. The trays were left open in a laminar flow hood for 3 h to allow evaporation of the excess water on the leaf surface. Mycelial agar plugs (5 mm in diameter) of *S. sclerotiorum* were inoculated on the leaves with the mycelia facing the leaves, two mycelial agar plugs on each leaf and four leaves for each treatment. The trays were individually covered with transparent plastic films (0.1 mm thick) to maintain high humidity and placed in a growth chamber at 20 °C under fluorescent light (12 h per day) for 2 d. The lesion diameter around each mycelial agar plug was measured. The assays were repeated three times.

### 2.8. Detection of Endophytic Growth

Seeds of oilseed rape “Zhongyou No. 9” were pre-germinated for 4 d, and then, they were sown in the moisturized Organic Culture Mix in seedling trays, which were maintained for 10 d in the growth room (20 °C). Then, the seedlings were drenched with the conidial suspension of JC-1 (1 × 10^7^ conidia/mL) or water alone (control), 20 mL per seedling. After incubation for another 30 days, six seedlings for each treatment were uprooted and washed under running water to remove the culture mix. The seedlings were blot-dried, the taproots and stems were cut into segments of 1 to 2 mm long, and the true leaves were cut into small square-shaped pieces (~5 × 5 mm, length × width). The segments of taproots/stems and the leaf pieces were surface-sterilized with 75% ethanol (30 s) and 5% NaClO (2 min). After washing in sterilized water (three times, for 30 s each time), the plant tissues were placed on PDA amended with lactic acid to inhibit bacterial contamination, and the cultures were incubated at 20 °C for 20 d. Colonies of JC-1 formed from the plant tissues were identified based on the particular morphological features of this fungus. The number of segments of taproots/stems and the leaf pieces with the emergence of JC-1 was recorded.

### 2.9. Induction of Systemic Resistance

Seeds of the cultivar “Zhongyou No. 9” were surface-sterilized with NaClO, followed by immersing in the conidial suspension of JC-1 (1 × 10^7^ conidia/mL) or water alone (control) for 12 h (Appendix A). They were then transferred to moisturized filter papers in Petri dishes, which were placed for 4 d in a growth room (20 °C, 16 h of light/8 h of darkness). The germinated seeds of both treatments were sown in the moisturized Organic Culture Mix in 32-hole seedling trays, one seed in each hole. The trays were placed in the growth room for 4 d, and the seedlings were drenched with the conidial suspension of JC-1 or water alone, 10 mL per seedling. Eight days later, the cotyledons were wounded on both sides of the main veins using a sharp needle. Aliquots of the conidial suspension of *L. biglobosa* W10 (1 × 10^7^ conidia/mL) were pipetted onto the cotyledons, 10 μL on each wound, 14 cotyledons on seven seedlings for each treatment. The treated seedlings were incubated at 20 °C for 8 d under humid conditions (R.H. > 90%). The diameter of the necrotic lesion around each wound was measured. The assay was repeated three times as three independent trials.

### 2.10. Quantification of Anthocyanins

Seeds of oilseed rape “Zhongyou No. 9” were treated with the conidial suspensions of JC-1 (1 × 10^5–7^ conidia/mL) or water alone (control), and they were separately placed on moisturized filter papers for 4 d (20 °C). The germinated seeds of each treatment were selected, and the hypocotyl together with the cotyledons on each seedling were removed. The plant samples with the hypocotyls and cotyledons (0.3 g) for different treatments were separately transferred to 2-mL-centrifuge tubes. Aliquots of the HCl-acidified methanol were added to the tubes (1 mL per tube) to extract the anthocyanins from the plant samples by shaking the tubes on a rotary shaker (70 rpm) at room temperature for 18 h [46]. The mixtures in the tubes were centrifuged at room temperature at 12,000 rpm, the supernatants (0.4 mL) were pipetted out and transferred to other 2-mL tubes, and aliquots (0.6 mL) of the acidified methanol were added. The optical density (OD) values of the resulting mixtures were measured at wavelengths of 530 nm and 657 nm in a spectrophotometer. The content of anthocyanins (*CA*) was calculated with the following formula:$$\mathrm{CA}\;(\mathrm{nmol}/g)=\frac{OD530-0.25\times OD657}{\epsilon }\;\times \;\frac{v}{m}\;\times 1,000,000$$
where *OD530* and *OD657* represent the *OD* values at 530 nm and 657 nm, respectively, ϵ is a constant (4.62 × 10^6^), *v* represents the volume of the mixture for measurement (1 mL), *m* represents the weight of each plant sample (0.3 g). The measurement was repeated three times.

### 2.11. Quantitative Detection of Expression of the Defense-Related Genes

Seeds of oilseed rape “Zhongyou No. 9” were surface-sterilized and immersed for 12 h in the conidial suspensions of JC-1 (1 × 10^5–7^ conidia/mL) or in water alone (control), and they were then transferred to moisturized paper filters in Petri dishes. After incubation for 4 d (20 °C), the hypocotyls and radicles (0.7 g) for each treatment were sampled and ground into a fine powder in liquid nitrogen. Total RNA was extracted from the powder using the PLANTpure Plant RNA Kit (Aidlab Biotechnol. Co., Ltd., Beijing, China), and DNA in the extract was eliminated using RNAase-free DNase I (Appendix A). The RNA was then used as a template for the synthesis of cDNA under the catalysis of PrimeScript II RTase. Finally, the cDNA was used as a template to quantify the transcripts of *CHI* (for chalcone isomerase), *NCED3* (for 9-cis-epoxycarotenoid dioxygenase), *PR1* (for pathogenesis-related protein 1) and *ACT7* (for actin, reference) in quantitative RT-PCR (qRT-PCR) using the primer pairs CHI/CHIR, NCED3F/NCED3R, PR-1F/PR-1R and ACT7F/ACT7R (Appendix A), respectively. The transcript level of each gene was calculated using the ^ΔΔ^Ct method [47]. For normalization of the data, the transcript level of each gene in the control treatment was considered to be 1.0, and the scale was used to calibrate the transcript level of that gene in the treatments of JC-1. The qRT-PCR was repeated independently three times.

### 2.12. Data Analysis

Data on percentages of inhibition of mycelial growth, leaf lesion diameters, content of anthocyanins, as well as the relative expression levels of *CHI*, *NCED3* and *PR1*, was analyzed using the procedure of analysis of variance (ANOVA) in the SAS software (SAS Institute, Cary, NC, USA, v. 8.0, 1999). The means for different treatments in an experiment or a trial were separated using the least significant difference (LSD) test at α = 0.05. Before ANOVA, the percentage data was arcsine-transformed to angular values, and after analysis, the average angular values were individually back-transformed to numerical percentages. Data on cotyledon lesion diameters caused by *L. biglobiosa* in the resistance-inducing trials were analyzed using the procedure of univariate analysis in SAS. The means for the treatments of JC-1 and control in each trial were compared using a Student’s *t* test at α = 0.05, 0.01 or 0.001.

## 3. Results

### 3.1. Taxonomic Identity

The JC-1 strain formed white floccose colonies on PDA at 20 °C with a villiform appearance on the surface (Figure 1A) and a pale yellow color on the colony back. Conidiophores (phialides) arose from the hyphae and tapered towards the apex; they are hyaline, aseptate, 12 to 30 μm long (averagely 23 μm long), mostly solitary, and some in pairs or in simple whorls. Globose to subglobose conidial masses (heads) were produced at the apex of the conidiophores (Figure 1B). Conidia are unicellular, hyaline, oval in shape, sharp at both ends, and 3.3–4.5 μm × 2.1–3.3 μm in size (Figure 1C). These morphological characteristics matched descriptions for the fungi in the genus *Simplicillium* [1], suggesting that the JC-1 strain belongs to a species of *Simplicillium*.

The partial DNA sequences of ITS, LSU, SSU and *TEF1α* in JC-1 were obtained by PCR, and they were submitted to GenBank and assigned with the GenBank accession numbers: MT807906, MT807907, MT807908 and MT826785, respectively (Table 1). Phylogenetic analysis based on ITS alone and the concatenated sequence of ITS/LSU/SSU/*TEF1α* showed that JC-1 is closely related to *S. lamellicola* CBS 116.25^T^ and CBS 454.70 (Figure 2 and Appendix A). Therefore, the JC-1 strain belongs to *Simplicillium lamellicola* (Smith) Zare & Gams.

### 3.2. The Genome of JC-1 and the Gene Clusters for the Secondary Metabolites

The genome of *S. lamellicola* JC-1 was sequenced, the PacBio long reads (12.47 Gb) were used for genome assembly, and the MGISEQ-2000 and Illumina short reads (3.5 and 3.1 Gb, respectively) were used for correction of the genome. Finally, a genome was assembled on 22 contigs with a total length of 29.14 Mb (Table 2), including 28.98 Mb on 10 major contigs accounting for 99.45% of the total genome (Figure 3) and 0.16 Mb on 12 minor contigs accounting for 0.55% of the total genome. A total of 10,488 protein-coding genes were predicted in the genome of JC-1 (Table 2). The genome data was deposited in the China National GeneBank (CNGB) DataBase with the accession number CNP0003490.

Results from the antiSMASH search identified 37 gene clusters for the biosynthesis of the secondary metabolites or SMs (Appendix A). They were grouped into eight biosynthetic types, including NRPS/NRPS-like/NPRS-PKS (21), PKS (8), terpene (4), β-lactone (1), NAPAA/PKS (1), PKS/indole (1), and RiPP/terpene (1) (Appendix A). Eight out of the 37 putative SMs are known metabolites, including aureobasidin A1, duclauxin, neurosporin A, nivalenol, phomasetin, squalestatin S1, verlamelin and wortmanamide A (Appendix A), and the remaining 29 SMs are unknown metabolites. Three of the known SMs (e.g., aureobasidin A1, squalestatin S1 and verlamelin) have been reported to have antifungal activity [27,48,49]. The previously-characterized biosynthetic gene clusters (BGCs) for these three SMs [27,50,51] were used as references for identification of the homologous SMs in the genome of JC-1. The BGC for aureobasidin A1 was detected in Contig No. 1 (Appendix A, Appendix A), and the amino acids encoded by the backbone gene (Sl000517) are 52% identical to those of the aureobasidin A1 biosynthesis complex (*aba1*) in *Aureobasidium pullulans* [51]. The BGC for verlamelin was detected in Contig No. 1 and No. 4, and it accommodates four genes, including Sl000045, Sl000044 and Sl000046 in Contig No. 1 as well as Sl005221 in Contig No. 4. These four genes are homologous to *vlmS*, *vlmB*, *vlmC* and *vlmA*, respectively, in *Lecanicillium* HF627 [29]. The nucleotides of Sl005221, Sl000045, Sl000044 and Sl000046 are identical by 51%, 92%, 90% and 90% to those of *vlmA*, *vlmB*, *vlmC* and *vlmS*, respectively. The amino acids encoded by Sl000045, Sl000044 and Sl000046 are identical by 47%, 96%, 96% and 92% to those encoded by *vlmA*, *vlmB*, *vlmC* and *vlmS*, respectively (Appendix A). The BGC for squalestatin S1 in JC-1 was detected in Contig No. 4 (Appendix A, Appendix A), and the backbone gene (Sl005901) codes for the amino acids with 58% identity to that of the squalene synthetase encoded by *mfR6* in *Phoma* MF5453, a well-characterized producer of squalestatin S1 [50].

Transcriptome profiling showed that in the 3- and 9-day-old PDB cultures, the backbone genes for the eight known BGCs had expression, although they varied greatly in expression level as evaluated by FPKM values (Figure 4). The backbone gene Sl005901 for squalestatin S1 is highly expressed, with a FPKM higher than 100, followed by the backbone gene Sl003510 for phomasetin, with an average FPKM of 41. On the other hand, the backbone gene Sl005538 for wortmanamide A had negligible expression, with a FPKM lower than 1.0. The verlamelin backbone gene Sl000046 and the aureobasidin A1 backbone gene Sl000517 had FPKM values ranging from 1.4 to 4.0 (Figure 4).

### 3.3. Antifungal Activity

Results of the dual culturing test showed that the JC-1 strain strongly inhibited the mycelial growth of *B. cinerea*, *L. biglobosa* and *S. sclerotiorum*, as indicated by the formation of inhibition zones (clear zones) between the colonies of JC-1 and these three target fungi in the dual cultures (Figure 5). The average width of the inhibition zones (*n* = 3) was 1.6, 1.7 and 1.5 cm between JC-1 and *L. biglobosa*, JC-1 and *B. cinerea*, and JC-1 and *S. sclerotiorum*, respectively.

The antifungal metabolites in the PDB cultures were extracted with acetone and ethyl acetate (Appendix A). The crude extract showed antifungal activity against *S. sclerotiorum*; the percentage of mycelial growth inhibition increased from 24% to 92% when the concentration of the crude extract was increased from 12.5 to 800 μg/mL (Figure 6C). The half-effective concentration (EC_50_) of the crude extract was 33.2 μg/mL.

### 3.4. Control Efficacy of the Cultures and the Crude Extract

In the supernatant trial, the leaves of oilseed rape treated with water alone formed large necrotic lesions around the mycelial agar plugs of *S. sclerotiorum* (20 °C, 2 d), the average lesion diameter reached 1.8 cm. However, the leaves of oilseed rape treated with the supernatants from the 9-, 12- and 15-day-old cultures of JC-1 (SU-9d, SU-12d and SU-15d, respectively) produced no visible lesions (Figure 7A). In the crude extract trial, while the control treatment (CK) also formed large necrotic lesions with the average lesion diameter at 2.3 cm and the fungicide treatment (e.g., PR-450) formed no visible lesions, the three treatments of the crude extract (e.g., CE-200, CE-400 and CE-800) formed lesions with average diameters ranging from 0.3 to 2.4 cm (Figure 7B). While the treatment with a low concentration of the crude extract (e.g., CE-200) produced large lesions with an average diameter of 2.4 cm, which had no significant difference (*p* > 0.05) from that in the control treatment, the treatments with high concentrations of the crude extract (e.g., CE-400, CE-800) produced small lesions with average diameters of 1.0 and 0.3 cm, respectively, significantly lower (*p* < 0.05) than that in the control treatment. Interestingly, the average lesion diameter in CE-800 had no significant difference (*p* > 0.05) from that in the treatment of SU-15d (0.3 cm in diameter), suggesting that the crude extract at 800 μg/mL had a control efficacy equal to that of the 15-day-old cultures.

### 3.5. Endophytic Growth and Induced Systemic Resistance

Seeds of oilseed rape treated with JC-1 (1 × 10^7^ conidia/mL) or water alone for 12 h or 24 h germinated after incubation at 20 °C for 4 d, the percentages of the germinated seeds reached 60%–80%, and no significant difference (*p >* 0.05) was detected between the treatments of JC-1 and water alone (Appendix A). However, the seedlings of the two treatments differed significantly (*p* < 0.05) in length of radicals (roots) and hypocotyls (shoots), the seedlings in the treatment of JC-1 formed shorter hypocotyls than those in the control treatment (Appendix A); however, they produced longer radicals than those in the control treatment (Appendix A). Moreover, the results from the seedling-drenched inoculation experiment showed that JC-1 existed in the stems and leaves of oilseed rape without causing any visible disease symptoms; 23.8% of stem segments and 14.3% of leaf pieces were detected as positive for *S. lamellicola*; however, the root segments were detected negative for *S. lamellicola* (Table 3).

Results from the cotyledon-inoculation trials showed that necrotic lesions caused by *L. biglobosa* appeared on the cotyledons of the seedlings drenched with the conidial suspension of JC-1 (1 × 10^7^ conidia/mL) and water alone. Statistical analysis indicated that drenching with the JC-1 conidia formed significantly (*p* < 0.001) smaller cotyledon lesions than drenching with water alone (Figure 8). The average cotyledon lesion diameters in the treatment of drenching with JC-1 were reduced by 22%, 16% and 32% in trials 1, 2 and 3, respectively, compared to those in the control treatment of drenching with water alone.

### 3.6. Mechanisms for the Enhanced Systemic Resistance

The seedlings of oilseed rape from the seeds treated with the JC-1 conidial suspensions (1 × 10^5–7^ conidia/mL) and water alone (control) had accumulations of anthocyanins. The treatments of JC-1 at 1 × 10^5^, 1 × 10^6^ and 1 × 10^7^ conidia/mL had the content of anthocyanins at 56.7, 70.8 and 67.9 nmol/g, respectively, and the values were significantly (*p* < 0.05) higher than that of 49.5 nmol/g in the control treatment (Figure 9A).

Results of qRT-PCR showed that the three defense-related genes, *CHI*, *PR1* and *NECD3*, expressed in the seedlings treated with JC-1 (1 × 10^5–7^ conidia/mL) and water alone (control). An up-regulated expression of *CHI* and *PR1* was observed in the seedlings treated with JC-1 at 1 × 10^5^ conidia/mL (5 and 15 folds of the control, respectively) and 1 × 10^6^ conidia/mL (7 and 19 folds of the control, respectively). However, the seedlings treated with JC-1 at 1 × 10^7^ conidia/mL had relative expression values of the two genes similar to those in the control treatment (Figure 9B,C). On the contrary, the seedlings of oilseed rape showed a different expression pattern for *NECD3* from that of *CHI* and *PR1* in response to JC-1 (Figure 9D). In the treatment with JC-1 at 1 × 10^5^ conidia/mL, the seedlings had a relative expression level of *NECD3* similar to that in the control treatment. However, in the treatments with JC-1 at 1 × 10^6^ and 1 × 10^7^ conidia/mL, the seedlings showed a down-regulated expression of *NECD3*, as the relative expression values of *NECD3* in these two treatments of JC-1 were significantly (*p* < 0.05) decreased by 10% and 91%, respectively, compared to that in the control treatment.

## 4. Discussion

This study revealed that the JC-1 strain belongs to *Simplicillium lamellicola* (F. E. W. Smith) Zare & W. Gams. According to the previous record, *S. lamellicola* is a common fungus with a world-wide distribution, and it has been reported to be able to infect mushrooms like *Agaricus biaporus* and *A. bitorquis*, causing gill mildew and small brown spots, respectively [1]. Moreover, *S. lamellicola* has the capability of living with rust fungi as a mycoparasite as well as living with cysts of *Heterodera glycines* and eggs of *Meloidogyne arenaria* as a nematode parasite [1]. The JC-1 strain of *S. lamellicola* was isolated from a diseased pod of oilseed rape infested with *L. biglobosa.* The re-inoculation experiment indicated that *S. lamellicola* JC-1 hardly infected the cotyledons of oilseed rape as *L. biglobosa* did. Therefore, *S. lamellicola* JC-1 is not a pathogen of oilseed rape (Li, 2019). Moreover, mixed inoculation of the conidia of *S. lamellicola* (*Sl*) and *L. biglobosa* (*Lb*) on oilseed rape (even at a ratio as high as 8:2, *Sl:Lb*) still caused large necrotic lesions of similar sizes to those inoculated with *L. biglobosa* alone [52]. This result suggests that *S. lamellicola* JC-1 could not effectively inhibit infection by *L. biglobosa* through physical contact, implying that it is at least not a specific and highly-aggressive mycoparasite of *L*. *biglobosa.*

Zhang et al. (2014) obtained three strains of *S. lamellicola* from healthy plants of oilseed rape (*Brassica napus*), and these strains were considered endophytes of oilseed rape, although the endophytic lifestyle of these strains was not confirmed by re-inoculation and re-isolation [30]. In this study, we re-inoculated *S. lamellicola* JC-1 on the roots of oilseed rape; fortunately, it was re-isolated from the leaves and stems of the inoculated seedlings without showing any visible disease symptoms. This result suggests that *S. lamellicola* JC-1 is really an endophyte of oilseed rape. Further study to trace the infection and spread of *S. lamellicola* JC-1 in the tissues of oilseed rape using certain marker genes such as the green fluorescent protein gene (*gfp*) is warranted.

The present study demonstrated that seed/root-inoculation with the conidial suspension of *S. lamellicola* JC-1 caused a significant (*p* < 0.001) reduction in cotyledon size of lesions caused by *L. biglobosa* on the resulting seedlings of oilseed rape. This result indicated that *S. lamellicola* JC-1 could induce systemic resistance (ISR) against infection by *L. biglobosa*. The possible mechanisms underlying the ISR may lie in endophytic growth in plant tissues of oilseed rape and the consequent enhanced accumulation of anthocyanins, as well as up-regulated expression of *CHI* and *PR1*. It is well recognized that the PR1 gene is associated with the salicylic acid (SA) pathway, and up-regulated expression of the SA pathway can activate the biosynthesis of some phenol compounds and analogs, such as anthocyanins, phytoalexins and lignin, which act as physical and physiological barriers to defend against pathogen infection [53]. On the other hand, the NCED3 gene is associated with the biosynthesis of abscisic acid (ABA), and up-regulation of the ABA pathway usually accelerates plant senescence. Lowe et al. (2014) reported that *L. biglobosa* infects oilseed rape in a necrotrophic way [54], and down-regulation of expression of *NCED3* may lower the ABA level, thereby enhancing resistance against infection by *L. biglobosa* possibly through slowing down the senescence of the cotyledons of oilseed rape. This result also suggests that *S. lamellicola* JC-1 can be potentially used as a seed coating agent to suppress infection by soilborne pathogens such as *Plasmodiophora brassicae* (clubroot), *Rhizoctonia solani* (damping-off), as well as some airborne pathogens such as *Peronospora parasitica* (powdery mildew). Further field evaluation of the biocontrol efficacy of seed coating with *S. lamellicola* JC-1 against these diseases of oilseed rape is needed.

Kim et al. (2002) reported that *S. lamellicola* is a strong antagonist with the capability of producing the antifungal cyclic lipopeptide verlamelin [27]. A similar result was observed in this study, *S. lamellicola* JC-1 displayed strong antifungal activity against *B. cinerea*, *L. biglobosa* and *S. sclerotiorum*. So far, the antifungal metabolites produced by *S. lamellicola* JC-1 were not purified and identified. As an alternative, we analyzed the genome of *S. lamellicola* JC-1 to elucidate its capability of producing antifungal metabolites. We found that this particular fungus can produce verlamelin, aureobasidin A1 and squalestatin S1, which might be at least partially responsible for the antifungal activity of *S. lamellicola* JC-1. Moreover, genome analysis revealed that *S. lamellicola* JC-1 has the capability of producing 29 unknown metabolites, suggesting that *S. lamellicola* JC-1 is a prolific producer of antifungal metabolites. Further purification and identification are worthwhile to clarify the known and novel metabolites produced by JC-1.

Based on the reference biosynthetic gene cluster (BGC) for verlamelin in *Lecanicillium* HF627 [29], this study identified four genes (Sl000044, Sl000045, Sl000046 and Sl005221) putative for biosynthesis of verlamelin in *S. lamellicola* JC-1. Three (Sl000045, Sl000044 and Sl000046) of the four genes are clustered in Contig. No. 1, and they are highly identical to *vlmB*, *vlmC* and *vlmS*, respectively (≥90%) in *Lecanicillium* HF627 [29]. However, the other gene (Sl005221) locates in Contig No. 4 with a low identity (51% by nucleotides) to *vlmA* in *Lecanicillium* HF627 [29]. This comparison suggests that the verlamelin BGC in *S. lamellicola* JC-1 differs from that in *Lecanicillium* HF627, as Sl005221 is physically separated from Sl000045, Sl000044 and Sl000046 in *S. lamellicola* JC-1, whereas the four verlamelin BGC genes (*vlmA*, *vlmB*, *vlmC* and *vlmS*) in *Lecanicillium* HF627 are tightly clustered [29]. Further study is worthwhile to characterize the biosynthesis of verlamelin in *S. lamellicola* JC-1 using gene manipulation techniques (e.g., gene knockout and complementation).

The cultural filtrates of *S. lamellicola* JC-1 showed a high efficacy in suppressing infection by *S. sclerotiorum* on leaves of oilseed rape. Moreover, the antifungal activity of the cultural filtrates was stable under heat (121 °C/30 min) and the extreme acidic (pH 2)/alkaline (pH 13) conditions, as well as under UV irradiation. These results suggest that the metabolites in the cultures of *S. lamellicola* JC-1 have the potential to be used by spray on air parts of plants for control of sclerotinia stem rot as well as other fungal diseases of oilseed rape. This study tried to extract the antifungal metabolites from the cultures of *S. lamellicola* JC-1, the crude extract at 800 μg/mL displayed the same control efficacy as the cultural filtrates did. This result indicated that the crude extract had an unexpectedly low control efficacy, possibly due to the loss of some important components in the process of extraction using acetone and ethyl acetate. Therefore, it is necessary to optimize the procedure to extract the active antifungal metabolites from the cultures of *S. lamellicola* JC-1 in the future, as this is very important for the formulation of the cultures of this fungus as a biocontrol agent.

In summary, this study revealed that the JC-1 strain belongs to *Simplicillium lamellicola*, it is a strong fungal antagonist through the production of antifungal metabolites, and that the filtrates of JC-1 in the PDB cultures showed a high efficacy in suppressing infection by *S. sclerotiorum* on leaves of oilseed rape. Genome analysis indicated that JC-1 owns the capability of producing multiple antifungal metabolites, including aureobasidin A1, squalestatin S1 and verlamelin. Inoculation of JC-1 on seeds of oilseed rape caused a suppressive effect on infection by *L. biglobosa* on the cotyledons of the resulting seedlings, suggesting that JC-1 can induce systemic resistance (ISR). Endophytic growth, accumulation of anthocyanins, up-regulated expression of *CHI* and *PR1*, and down-regulated expression of *NECD3* were found to be possibly responsible for the ISR. Therefore, *S*. *lamellicola* JC-1 is a promising and multifunctional biocontrol agent.

## Figures and Tables

**Figure 1 jof-09-00057-f001:**
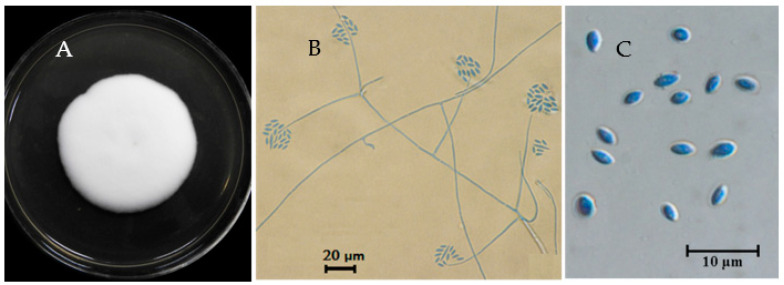
Morphology of *Simplicillium lamellicola* JC-1. (**A**) A colony with a white villiform appearance (PDA, 20 °C, 16 d) in a Petri dish (9 cm in diameter); (**B**) solitary and slender conidiophores (phialides) on hyphae with conidial masses at the apex (stained with cotton blue); (**C**) unicellular and oval-shaped conidia (stained with cotton blue).

**Figure 2 jof-09-00057-f002:**
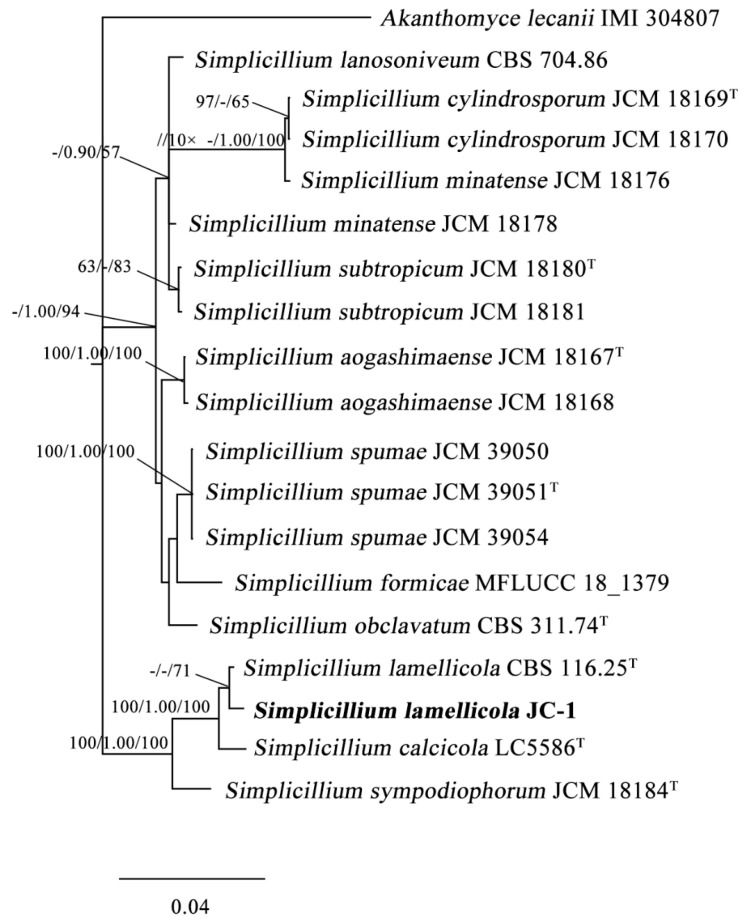
Phylogenetic tree of 19 fungal taxa in the genus *Simplicillium* and *Akanthomyces lecanii* (outgroup). The tree was constructed using concatenated sequences of ITS, LSU, SSU, and TEF1α (see Table 1 for GenBank accession numbers for these loci). RAxML bootstrap support values (ML) higher than 50, Bayesian posterior probability (PP) higher than 0.90, and maximum parsimony (MP) bootstrap support values higher than 50 were shown at the nodes (ML/PP/MP). The scale bar indicates sequence divergence. -, ML < 50 or PP < 0.90. Type strains were labeled with “T” at the top-right corner.

**Figure 3 jof-09-00057-f003:**
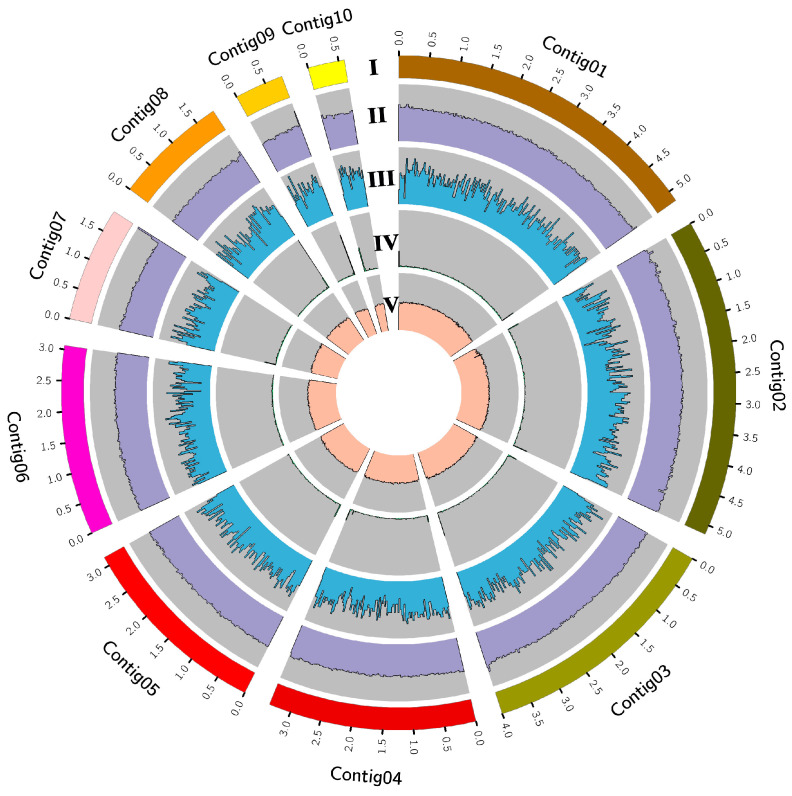
Circos-plot showing the genomic features of *Simplicillium lamellicola* JC-1. **I**, contigs larger than 0.5 Mb in length; **II**, frequency of distribution of reading coverage (scale: 0 to 200×); **III**, gene density (scale: 0 to 20); **IV**, repeat density (scale: 0 to 100%); and **V**, G + C content (scale: 0 to 100%). All statistics are based on non-overlapping windows (size = 30 kb).

**Figure 4 jof-09-00057-f004:**
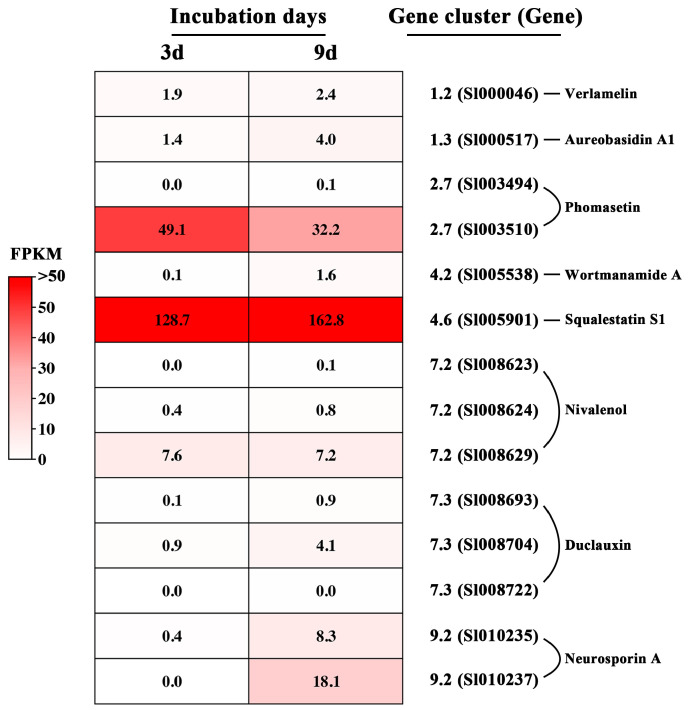
Heat map showing expression of the backbone genes for biosynthesis of eight known secondary metabolites in *Simplicillium lamellicola* JC-1 based on RNA profiling.

**Figure 5 jof-09-00057-f005:**
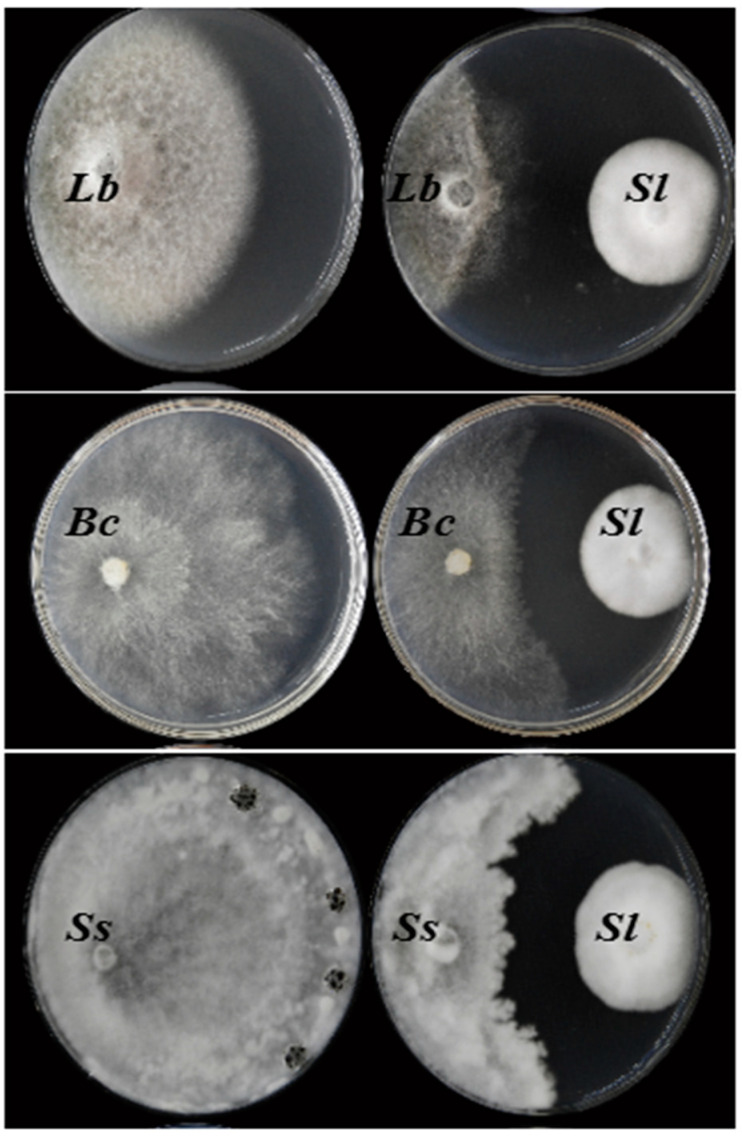
Dual cultures on potato dextrose agar showing the antagonistic interaction of *Simplicillium lamellicola* (*Sl*) JC-1 against *L. biglobosa* (*Lb*, 20 °C, 10 d), *B. cinerea* (*Bc*, 20 °C, 3 d) and *Sclerotinia sclerotiorum* (*Ss*, 20 °C, 3 d).

**Figure 6 jof-09-00057-f006:**
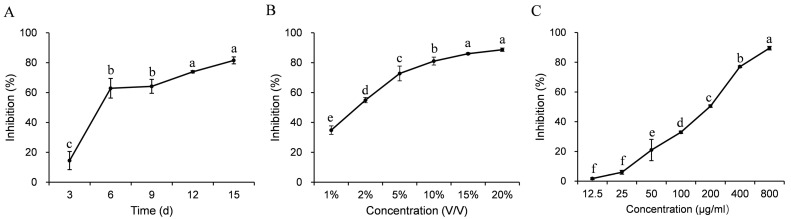
Antifungal activity of the cultures and the crude extract of *Simplicillium lamellicola* JC-1. The JC-1 strain of *S. lamellicola* was shake-incubated (20 °C, 150 rpm) in potato dextrose broth (PDB) for 3 to 15 d, and the cultures were centrifuged at 9000 rpm to obtain the supernatant. (**A**) Time-course of the antifungal activity of the 3- to 15-day-old DB cultures; (**B**) dose effect of the antifungal activity of the 15-day old PDB cultures; (**C**) antifungal activity of the crude extract of JC-1 from the 15-day-old PDB cultures. In each graph, means ± S.E. labeled with the same letters are not significantly different (*p >* 0.05) according to the least significant difference test.

**Figure 7 jof-09-00057-f007:**
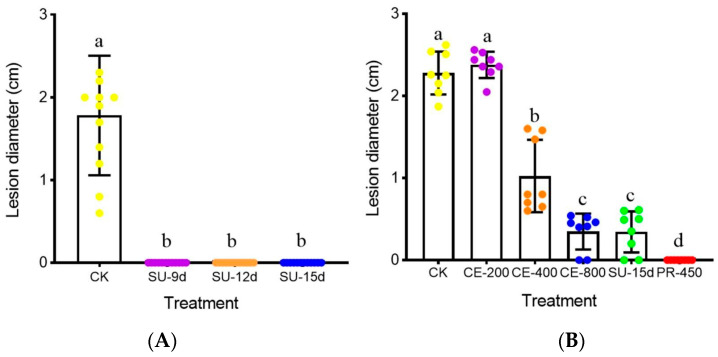
Efficacy of the cultures of *Simplicillium lamellicola* JC-1 and the crude extract in the suppression of infection by *Sclerotinia sclerotiorum* on detached leaves of oilseed rape (20 °C, 2 d). **CK**, control; **SU-9d**, **SU-12d** and **SU-15d**, supernatants from 9-, 12- and 15-day-old PDB cultures, respectively; (**A**) **CE-200**, **CE-400** and **CE-800**, the crude extract of JC-1 at 200, 400 and 800 μg/mL, respectively; **PR-450**, the fungicide prochloraz at 450 μg a.i./mL (**B**). In each histogram, bars (means ± S.E.) labeled with the same letters are not significantly different (*p >* 0.05) according to the least significant difference test.

**Figure 8 jof-09-00057-f008:**
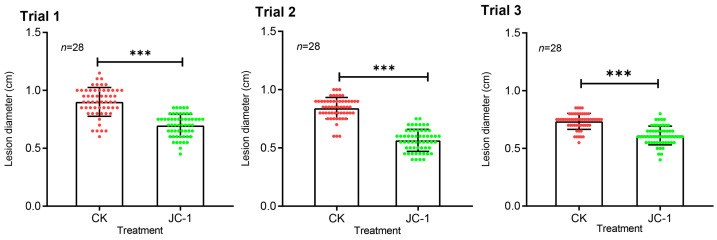
Histograms showing cotyledon lesion diameters on seedlings of oilseed rape treated with water alone (CK) and *S. lamellicola* (JC-1). The three trials were independently performed. ***, significantly different between CK and JC-1 at *p* < 0.001 according to a Student’s *t* test.

**Figure 9 jof-09-00057-f009:**
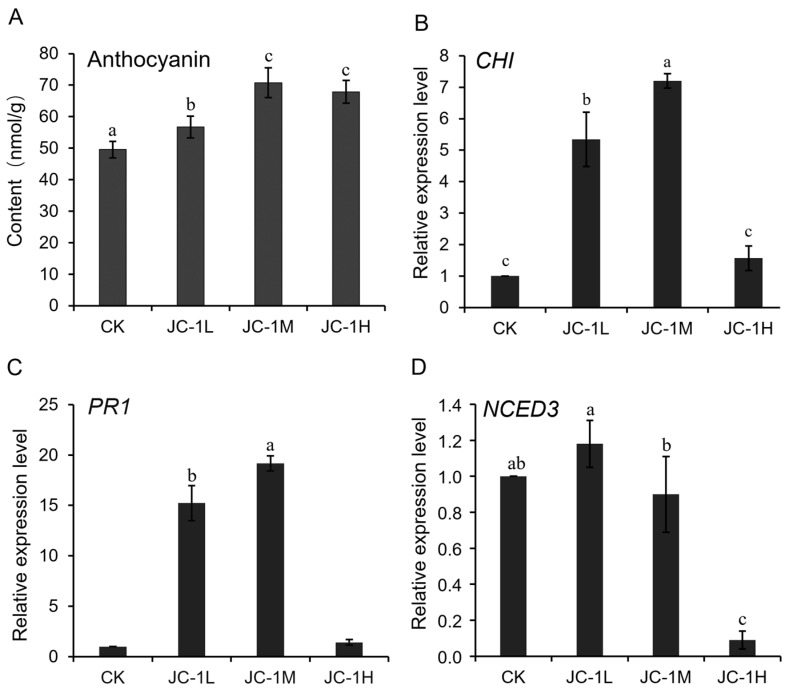
Content of anthocyanins (**A**) and relative expression levels of *CHI* (**B**) *PR1* (**C**) and *NECD3* (**D**) in the treatments of water alone (CK) as well as *S. lamellicola* at 1 × 10^5^ conidia/mL (Low conidial concentration or JC-1L), 1 × 10^6^ conidia/mL (Medium conidial concentration or JC-1M) and 1 × 10^7^ conidia/mL (High conidial concentration or JC-1H). In each graph, bars (means ± S.E., *n =* 3) labeled with the same letters are not significantly different (*p >* 0.05) according to the least significant difference test. *CHI*, *PR1* and *NECD3* code for chalcone isomerase, pathogenesis-related protein 1 and 9-cis-epoxycarotenoid dioxygenase, respectively.

**Table 1 jof-09-00057-t001:** Strains and the selected genes used in the phylogenetic analysis.

Species	Collection No.	Host or Niche	Country or Region	GenBank Acc. No.			
				ITS	SSU	LSU	*TEF1a*
*Akanthomyces lecanii*	CBS 101,247 = IMI 304807	*Coccus viridis*	West Indies	AJ292382	AF339604	AF339555	DQ522359
*S. aogashimaense*	JCM 18167	Soil	Japan	AB604002	LC496889	LC496874	LC496904
*S. aogashimaense*	JCM 18168	Soil	Japan	AB604004	LC496890	LC496875	LC496905
*S. calcicola*	LC5586 = CGMCC3.17943	Calcaire	China	KU746706	KY883301	KU746752	KX855252
*S. chinense*	LC 1342	Freshwater	China	JQ410323	-	JQ410321	-
*S. chinense*	LC 1345	Freshwater	China	JQ410324	-	JQ410322	-
*S. coffeanum*	CDA 734	*Coffea arabica*	Brazil	MF066034	-	MF066032	-
*S. cylindrosporum*	JCM 18169	Soil	Japan	AB603989	LC496891	LC496876	LC496906
*S. cylindrosporum*	JCM 18170	Soil	Japan	AB603994	LC496892	LC496877	LC496907
*S. filiforme*	URM 7918	*Citrullus lanatus*	Brazil	MH979338	-	MH979399	-
*S. formicae*	MFLUCC 18-1379	*Formicidae*	Thailand	NR_168789	NG_070121	NG_068624	MK926451
** *S. lamellicola* **	**JC-1**	** *Brassica napus* **	**China**	**MT807906**	**MT807908**	**MT807907**	**MT826785**
*S. lamellicola*	CBS 116.25 ^T^	*Agaricus bisporus*	UK	MH854806	AF339601	AF339552	DQ522356
*S. lamellicola*	CBS 454.70	-	USA	MH859793	-	MH871559	-
*S. lanosoniveum*	CBS 704.86	*Hemileia vastatrix*	Venezuela	AJ292396	AF339602	AF339553	DQ522358
*S. lanosoniveum* var. *tianjinienss*	CGMCC 4460	Blue-green alga	China	HM989951	-	-	-
*S. minatense*	JCM 18176	Soil	Japan	AB603992	LC496893	LC496878	LC496908
*S. minatense*	JCM 18178	Soil	Japan	AB603993	LC496894	LC496879	LC496909
*S. obclavatum*	CBS 311.74	Air	India	AJ292394	AF339567	AF339517	EF468798
*S. obclavatum*	JCM 18179	Soil	Japan	AB604000	-	-	-
*S. spumae*	JCM 39050	Foam	Japan	LC496869	LC496898	LC496883	LC496913
*S. spumae*	JCM 39051	Foam	Japan	LC496870	LC496899	LC496884	LC496914
*S. spumae*	JCM 39054	Foam	Japan	LC496871	LC496902	LC496887	LC496917
*S. subtropicum*	JCM 18180	Soil	Japan	AB603990	LC496895	LC496880	LC496910
*S. subtropicum*	JCM 18181	Soil	Japan	AB603995	LC496896	LC496881	LC496911
*S. sympodiophorum*	JCM 18184	Soil	Japan	AB604003	LC496897	LC496882	LC496912

Note: ITS = internal transcribed spacer (ITS1-5.8S rDNA-ITS2), SSU = small subunit ribosomal RNA gene, LSU = large subunit ribosomal RNA gene, *TEF1α* = translation elongation factor 1, -, not available.

**Table 2 jof-09-00057-t002:** Summary statistics of the *Simplicillium lamellicola* the JC-1 strain genome.

Variable	Statistics
Genome size	29.14 Mb
Genome coverage	427.93×
Number of scaffolds	22
Average contig length	1.32 Mb
Length of the largest contig	5.180 Mb
N_50_	3.35 Mb
GC content	48.37%
Protein coding genes	10,488
Gene clusters for secondary metabolites (SM)	38
Known SM gene clusters	8

Note: The genome data was deposited in the China National GeneBank (CNGB) DataBase with the accession number CNP0003490.

**Table 3 jof-09-00057-t003:** Number and percentages of tissues of oilseed rape (roots, stems and leaves) colonized by *S. lamellicola* JC-1.

Treatment ^1^	No. Tissues Colonized by *S. lamellicola*/Total Tissues (Percentage)		
	Root Segments	Stem Segments	Leaf Pieces
Water alone	0/21 (0)	0/21 (0)	0/21 (0)
Conidial suspension of JC-1	0/21 (0)	5/21 (23.8%)	3/21 (14.3%)

^1^ Ten-day-old seedlings of oilseed rape were inoculated with the conidial suspension of the JC-1 strain or with water alone. The seedlings were incubated for another 30 days, and then, colonization of roots, stems and leaves by *S. lamellicola* was then detected by tissue isolation on PDA.

## Data Availability

The sequences of the contigs together with the raw sequencing data for *S*. *lamellicola* JC-1 were submitted to the China National GeneBank (CNGB) DataBase with the accession number CNP0003490.

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
