# Peer review of "Antifungal Activity and Biocontrol Potential of Simplicillium lamellicola JC-1 against Multiple Fungal Pathogens of Oilseed Rape"

_jof, 2022, doi:10.3390/jof9010057_

Round 1

Reviewer 1 Report

This paper appears sound and of interest to people like myself who are interested in control of oil crop diseases via genetic resistance but who are not directly concenrned by chemicals. For me, it appears satisfactory for publication, although I do not know whether it provides original results compared with previous publications.  

Author Response

Thank you for your comments. This article has several innovations, such as: 

  1. This paper first verified that Simplicillium lamellicola can colonize in oilseed rape as an endophyte.
  2. We found that that Simplicidium lamellicola has antifungal activity against Sclerotinia sclerotiorum and Leptocephalia biglobosa.
  3. We found that JC-1 has a distinct gene cluster for biosynthesis of verlamelin.

Reviewer 2 Report

The manuscript titled “Antifungal Activity and Biocontrol Potential of Simplicillium lamellicola JC-1 against Multiple Fungal Pathogens of Oilseed Rape” by Li et al. describes exploration of plant beneficial properties of endophytic fungal isolate JC-1 from oilseed rape, use both morphological and molecular characteristics to identify the isolate as Simplicillium lamellicola, antifungal activity of its metabolites by means of bioassays and genome-based analysis, and its ability to induce plant defense response.

Overall, the manuscript describes comprehensive research work and is of high relevance for phylogeny and evolutionary relationships of genus Simplicillium, ecology and discovery of novel compounds for application in biocontrol.

Author Response

Thank you for your comments! We have carefully read the text and tried our best to correct the editorial mistakes. We think that quality of the manuscript has been greatly improved.

Reviewer 3 Report

In this MS, the authors isolated Simplicillium lamellicola from oilseed rape and studied the antifungal capacity of this strain. The work is well conducted and I only have a few suggestions.

- Include a section of conclusions.

- For further analysis, use MS-TOF or another chromatographic technique to identify antifungal compounds.

- Read the text carefully, I can detect typing error

Author Response

Thank you for your comments!

-Include a section of conclusions.

Response: Accepted. A paragraph for conclusions was added at the end of the “Discussion” part.

- For further analysis, use MS-TOF or another chromatographic technique to identify antifungal compounds.

Response: Accepted. This is a very good suggestion. We discussed that further identification of the antifungal compounds produced by strain JC-1 is worthwhile in the future studies using chemical purification and chromatographic techniques. However, it is pretty hard to include these studies in this paper. The reasons are: (i) the first author (Ms Wenting Li) graduated and is working in another unit, it is impossible to for her to undertake these tasks any more; (ii) The standard compounds for aureobasidin A1, squalestatin S1 and verlamelin are not available in the commercial chemical markets, and this situation affect our identification of the known compounds produced by JC-1; (iii) There are at least 29 biosynthetic gene clusters for known metabolites according to prediction by antiSMASH, and this result suggests that it may not be an easy thing to identify the antifungal metabolites of JC-1 solely using MS-TOF and chromatographic techniques. In stead, we plan to initiate a comprehensive study to isolate and identify the antifungal metabolites.  

- Read the text carefully, I can detect typing error

Response: Accepted. We have carefully read the text and tried our best to correct the editorial mistakes.